# Iliacus Muscle Hematoma an Uncommon Complication in a Rehabilitation Unit: A Case Report Study

**DOI:** 10.3390/healthcare10020297

**Published:** 2022-02-03

**Authors:** Riccardo Battaglia, Antonio Cerasa, Maria Elena Pugliese, Lucia Francesca Lucca, Paolo Tonin

**Affiliations:** 1Intensive Rehabilitation Unit, S’Anna Institute, 88900 Crotone, Italy; me.pugliese@isakr.it (M.E.P.); l.lucca@istitutosantanna.it (L.F.L.); patonin18@gmail.com (P.T.); 2Institute for Biomedical Research and Innovation (IRIB), National Research Council of Italy, 98164 Messina, Italy; 3Pharmacotechnology Documentation and Transfer Unit, Preclinical and Translational Pharmacology, Department of Pharmacy, Health Science and Nutrition, University of Calabria, 87036 Rende, Italy

**Keywords:** iliacus muscle hematoma, bed–wheelchair transfer, case report, neurorehabilitation ward

## Abstract

Iliacus muscle hematoma is a very rare condition in rehabilitation wards. A 35-year-old pregnant woman at seven months with cerebellar hematoma with signs of compression underwent a cesarean procedure together with neurosurgical decompression of the posterior cranial fossa procedure. After a period in an intensive unit without any complications, she was admitted to our neurorehabilitation ward and treated with antihypertensive therapy, heparin, and anti-seizure drugs. During a rehabilitation session, after a bed–wheelchair transfer, she was feeling very unwell suffering from severe right leg pain radiating to the mid-face leg. Despite unremarkable physical evidence (skin appearance, temperature, and peripheral arterial pulse in both legs), imaging data (ultrasound, CT, and MRI) confirmed the presence of a right iliac muscle hematoma (IMH). Enoxaparin was immediately stopped, and bed rest was prescribed in an antalgic position with tramadole acetaminophen for pain control. After a few days, the patient felt well and was discharged without any additional symptoms. Our study demonstrates that, when a sudden severe leg pain develops, IMH should be considered in the differential diagnosis. This finding suggests further research and tailored protocols for rehabilitation in patients at high risk for iliacus muscle hematoma.

## 1. Introduction

Iliacus muscle hematoma (IMH) is an unusual but potentially fatal event. It can be spontaneous or secondary to surgery procedures, trauma, and bleeding disorders. Several risk factors are associated with spontaneous IMH but are not always present. Multicentric retrospective studies in the Intensive Care Unit estimated a mortality rate from 30 up to 50% when diagnosed [1]. Potential risk factors for muscle hematoma are obesity, chronic kidney disease, older age, anti-platelets, and anticoagulant medications [1]. Indirect damage and non-traumatic shear stress of the muscles are also described in the literature as causal factors of muscle hematoma [2].

Generally, the presence of acute leg pain with groin irradiation and a reduction in hip range of motion without lower limbs asymmetry and skin or vascular abnormalities, are considered clinical signs of IMH in the differential diagnosis [3]. Other clinical abnormalities can result from a drop in hemoglobin level which should be combined with evidence provided by imaging analysis, such as computerized tomography (CT) or magnetic resonance imaging (MRI). Fast ultrasound examination may also be helpful in ruling out other causes, such as gastroenteric and gynecological issues (appendicitis, abdominal/pelvic abscess with nerve involvement, or infiltrating tumor), vascular issues (venous thrombosis, peripheral artery disease, (PAD), or ischemic systemic embolization), or leg soft tissue/muscle issues (muscle tears, infection of muscles, or soft tissues) [4].

Clinical treatment is strictly dependent on symptoms and clinical status. An acute severe hemoglobin drop and shock are tackled with an endovascular embolization procedure, whereas an open surgery is reserved for a bailout approach. A conservative approach with risk factors correction is useful for those with a less severe clinical presentation, severe comorbidities and older age [5]. In the literature, the reports on pelvic muscle hematoma concern acute settings as emergency departments and intensive care units. Case reports in rehabilitative departments are lacking.

Here, we describe a case of a patient with IMH that occurred during the rehabilitation period to examine challenges facing clinical practice in this condition.

## 2. Case Report

A 35-year-old pregnant woman in the seventh month fainted at home during normal activity. She was immediately transferred to the nearest hospital, where a diagnosis of cerebellar hematoma with signs of compression was made. An urgent neurosurgery procedure for decompression of the posterior cranial fossa and a cesarean procedure for an early delivery was performed. She was admitted to the Intensive Care Unit (ICU) ward intubated and sedated. Endoscopic percutaneous gastrostomy together with a tracheostomy procedure was performed. She was gradually weaned by mechanical ventilation and awakened from sedative therapy. No complications were reported during the stay in ICU.

Past medical history was characterized by a poly-cystic ovarian syndrome treated with hormone therapy. Her home medication was enoxaparin prophylaxis during pregnancy. There was no brain hemorrhage or bleeding disorders in her family history. The patient had a normal diet, was a non-smoker, and no alcohol abuse or illicit drug was reported.

After 30 days, she was admitted to our neurorehabilitation unit. At first evaluation, she was in a minimally conscious state, showing minimal but definite, sustained, and reproducible behavioral evidence of awareness of self or environment (Coma Recovery Scale–revised (CRS-r) = 16). She breathed spontaneously through a tracheostomy cannula. Feeding and hydration were assured via gastrostomy, and a bladder catheter was present. There were not bedsore lesions. At admission, the patient was treated with antihypertensive therapy, low molecular weight heparin (enoxaparin), anti-seizure drugs prophylaxis, and protonic pump inhibitors. The patient underwent a rehabilitation program focused on passive mobilization of all the limbs in all range of motion by a therapist, sitting posture conditioning on tilting-chair with the use of device for the transfer, passive verticalization on the automated-bed, sensory stimulation and respiratory rehabilitation with positive airway pressure administration, and prono-supination to facilitate airways clearance and subsequent progressive closure of the tracheostomy cannula, helping the patient to regain the breathing trough nose, before a definitive cannula removal.

After a few weeks, the patient emerged from the minimally conscious state (CRS-r = 22), became more responsive to verbal stimuli, and was able to move arms and legs with severe cerebellar dysmetria. The tracheal tube was removed without complication and a swallowing therapy program was started. The bladder catheter was removed. The patient started an active program exercise on the bed with a therapist and a sitting posture conditioning with active participation of the patient. Her dysmetria was consistent and was unable to maintain a standing position without any advice in this phase, but she regained satisfying trunk control. The rehabilitation program was aimed at achieving complete autonomy of the patient, first in postural changes in bed and then in bed–wheelchair transfers. In the first phases of neuromotor recovery, the position from supine to sitting at the edge of the bed was reached with the help of the therapist, as well as the transfer from the bed to the wheelchair. After regaining control of the trunk in a sitting position on the edge of the bed with both feet resting on the ground, the patient was helped by the therapist to reach the upright position, rotate towards the wheelchair, and sit on it. In all these phases, the therapist helped the patient with both verbal suggestions and physical contact to obtain neuromotor control for safe autonomy of bed–wheelchair transfers.

However, during a bed–wheelchair transfer with the patient still being dysmetric but partially involved in the movement, she complained of severe right leg pain radiating to mid-face leg. She described the pain as 10/10 on the numerical rating scale (NRS) in passive and in active movement. At the bed, both legs were symmetric at the examination. The skin appearance and temperature were normal. Peripheral arterial pulse was present in both legs. The peripheral reflexes were reduced, but the patient was uncooperative due to the antalgic posture of the leg and severe discomfort. She was unable to move the right lower limb for intense pain at first degrees of range of motion. The passive hip extension was passible but evoked very intense pain. The pain was not responsive to nonsteroidal anti-inflammatory drugs. The only way to partially alleviate the pain was with a pillow under her knee to maintain the hip flex. The patient also complained of a mild tenderness at palpation of the inferior right abdomen. Cardiac and thoracic examinations were unremarkable. Her vital signs were normal with normal body temperature (BP 120/80 mmHg, HR 90 bpm, SpO2 98%, T 36.5 °C). Her blood test revealed an acute severe hemoglobin drop with level of 10.5 gr/dl (at admission 12.5 gr/dl), 39,500.0/mm^3^ platelets and 12,650/mm^3^ white blood cell count prevailing neutrophil, C-reactive protein 20.35 mg/dl (normal value < 0.5), creatinine 0.49 mg/dl, prothrombin time international normalized ratio (INR) 1.3 (normal value 0.80–1.20), activated partial thromboplastin time (aPTT) 1.14, and normal transaminases and creatine phosphokinase (CPK) levels. A right hip joint X-Ray was performed and revealed no abnormalities.

Three different instrumental imaging examinations were performed. An ultrasound study excluded a peritoneal effusion but was not more informative for extensive meteorism. CT without contrast revealed an enlargement of right iliac muscle, slightly parenchymal inhomogeneity without signs of overt bleeding and excluded other causes of leg pain (Figure 1).

Finally, an MRI examination definitively revealed an iliac right muscle enlargement, hypo and hyperintensity muscle lesions on T2 sequences better seen on fat attenuation sequences (STIR), and partially extending to gluteal muscles group. This is a pattern indicative of muscle hematoma with different lesions at different stages (Figure 2).

A second complete blood count revealed a stable hemoglobin level at 24 h. Due to normal vital signs and a stable hemoglobin level, a conservative approach was pursued. A surveillance echo-doppler leg vein was negative for signs of deep vein thrombosis. Enoxaparin was stopped and bed rest was prescribed in an antalgic position with tramadol e acetaminophen for pain control. After few days, the patient referred to a decrease in leg pain.

One week later, painkiller treatment was suspended, and a light bedridden physical therapy was begun to reduce muscle complication. At 2 weeks, she was able to stay in a wheelchair. At 3 weeks, she was able to start a rehabilitation program actively, with stand-up exercise, without any complaint. The exercises were set for her consciousness and cooperation level in relation to her pain alert level. At 1 month, she never complained of pain and continued her rehabilitation program with more intensive exercises for her hips. Clinical improvement was satisfactory, complete blood count was normal, and a second CT scan was unnecessary. She never received again enoxaparin during her stay without any consequence.

## 3. Discussion

Our data demonstrated that, when a sudden severe leg pain develops with preserved active and passive movements, antalgic position, and no vascular signs of leg asymmetry, an IMH should be considered in the differential diagnosis, as shown in previous clinical studies [6,7]. In the great majority of the literature, the onset of IMH is described in ICU settings, where this event can be more frequent. Our case study describes its onset in a rehabilitative setting. Excluding the genetic predisposition for bleeding disorders, among the common risk factors found in ICUs [8], the only one present in our patient was the use of anticoagulant [9], as standard prophylaxis in bedridden patients.

The causal event appears to be related to the stress of a de-conditioned muscle during the transfer from the bed to the wheelchair. The same causal event was reported by Okumura et al. [10] on a patient after total hip arthroplasty. However, in that case, the presence of surgical intervention, two days before the IMH onset, may justify a combination of risk factors, as suggested by the authors. Differently, in our report, only the standardized prophylaxis for bedridden patients was present. It implies, also, that common rehabilitation procedures such as wheelchair transfer may be more relevant in those at risk.

This evidence suggests that physicians, nurses, and therapists should be aware that this common procedure, if associated with the presence of anticoagulant therapy and other risk factors, could contribute to triggering IHM-related events which, in rare cases, could be life-threatening [9]. To avoid this kind of event, it could be useful to prescribe a tailored rehabilitation program for a certain period, including: (a) passive transfer to the wheelchair; (b) handling the limbs of the patient with a reduced range of motion to reduce muscle strain or overload; and (c) earlier suspension of anticoagulant for deep vein thrombosis prophylaxis or drugs predisposing for bleeding. All these procedures could be useful to avoid any muscle spontaneous hematoma such as that described in this case. Again, programmed blood sample analyses could also be performed to detect any variation useful to highlight similar high-risk conditions.

## 4. Conclusions

Our experience suggests that in the rehabilitation wards it would be advisable to set up good practice procedures to make the involved personnel aware of the possible risk and indicate the operating methods to reduce the occurrence of IMH. We encourage further research to identify patients and period at risk and to design a specific rehabilitation program.

## Figures and Tables

**Figure 1 healthcare-10-00297-f001:**
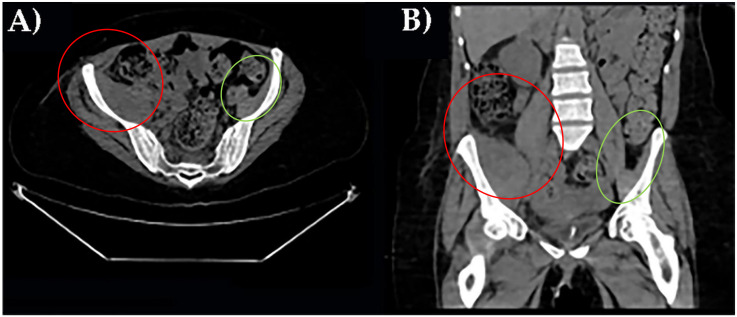
CT axial scan of the pelvis with right ileal muscle hematoma. (**A**). CT coronal scan of the pelvis with right ileal muscle hematoma (**B**). In red, pathological finding; in green, normal appearance.

**Figure 2 healthcare-10-00297-f002:**
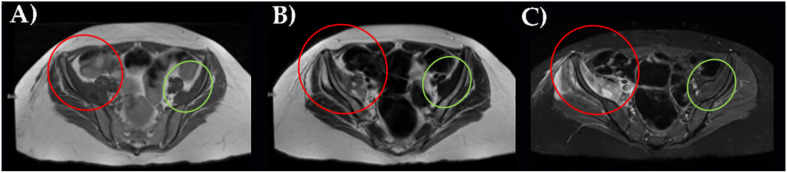
MRI of the pelvis with right ileal muscle hematoma performed with a T1- (**A**), T2- (**B**), and STIR sequences (**C**). In red, pathological finding; in green, normal appearance.

## Data Availability

Not applicable.

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
