# Peer review of "Iliacus Muscle Hematoma an Uncommon Complication in a Rehabilitation Unit: A Case Report Study"

_healthcare, 2022, doi:10.3390/healthcare10020297_

Round 1

Reviewer 1 Report

This is the reporting of a case of iliacus muscle hematoma in a 35-year-old woman on anticoagulant therapy undergoing neurorehabilitation. In general, this is a well written manuscript with good coverage on the clinical decision-making process. According to the authors, the most likely explanation for what have occurred was a combination of anticoagulant therapy in a deconditioned/structurally fragile muscle during bed to wheel-chair transfer. I feel, however, that details are missing to fully understand their explanation. Was bed-to-wheelchair transfer part daily life training activity, i.e., was the patient actively (muscle powered activity) participating in the transfer, or was it a passive action (assisted by a device or healthcare providers)? How is this performed in your rehabilitation unit? Also, the authors state that recommendations on transfer ought be developed but they do not point on directions. Do they think that, for example, a thorough physical examination on pelvic muscles strength/conditioning should be routinely performed before transfers from bed occur? What are their thoughts on that?

Specific comments:

Abstract

Lines 27–28: This recommendation is vague because the authors don’t discuss the procedures used during bed–chair transfer. Was this training or simply sitting someone on a wheelchair for staying in a more vertical position? Please, provide more information on that in your article.

Introduction

Line 35: Not sure what the authors mean by “causative event”. Traumatic? A single factor? Please, reword.

Line 47: “rulling out other causes”. Please provide some examples of other potential causes as you are discussing differential diagnosis regarding that clinical presentation.

Case report

Line 68: Not sure what the authors mean by “familiarity”. Do you mean “family history”?

Line 71: Please add the score of the CRS-r.

Line 76: Please add “(enoxaparin)” after “low molecular weight heparin”.

Line 79: Regarding “respiratory rehab” you already state “rehabilitation” at the beginning of the sentence. What aspects of respiratory rehab are relevant in this concern? Respiratory muscles training? Other? Please, be more specific. In the same line, what are the informed consent for? Rehab? This article?

Line 80: Please provide the new CRS-r score.

Lines 97-98: Many readers may not be familiar with the acronyms CPK, INR, aPTT, vn. Please provide the definitions.

Figures: Many readers may not easily detect abnormalities. Please, highlight them (suggestion: an arrow or a circle around the enlarged muscle)

Discussion:

Line 130: I’m a little bit confused with “preserved active and passive movements” because the authors earlier reported that “She was unable to move the right lower limb. The passive hip extension was limited and very painful” (lines 89–90). Can the authors please explain?

May physical examination predict such event reported in the article? A certain physical performance level should be previously achieved? Any recommendations/hypotheses?

Author Response

This is the reporting of a case of iliacus muscle hematoma in a 35-year-old woman on anticoagulant therapy undergoing neurorehabilitation. In general, this is a well written manuscript with good coverage on the clinical decision-making process. According to the authors, the most likely explanation for what have occurred was a combination of anticoagulant therapy in a deconditioned/structurally fragile muscle during bed to wheel-chair transfer. I feel, however, that details are missing to fully understand their explanation. Was bed-to-wheelchair transfer part daily life training activity, i.e., was the patient actively (muscle powered activity) participating in the transfer, or was it a passive action (assisted by a device or healthcare providers)? How is this performed in your rehabilitation unit? Also, the authors state that recommendations on transfer ought be developed but they do not point on directions. Do they think that, for example, a thorough physical examination on pelvic muscles strength/conditioning should be routinely performed before transfers from bed occur? What are their thoughts on that?

REPLY: The bed-wheelchair transfer was part of the patient routine program. We have now better specified this point during the rehabilitation period. We emphasize the bed-wheelchair movement because we hypothesize that this was a “stress movement” with a strong temporal correlation with pain onset. There are no physical examinations that can be routinely performed suggesting a muscle at risk, but we are able to identify patients at risk. A large amount of risk factors is recognized as causes of muscle hematoma and possible stress on a deconditioned muscle can cause it. We should focus on the possible window time risk for a tailored and cautious rehabilitation program. In this case, the patient actively participated in the transfer on the chair. Perhaps, this kind of active movement was made too early and done during anticoagulation therapy in a patient with motor deficits (i.e. dysmetria).

Specific comments:

Abstract

Lines 27–28: This recommendation is vague because the authors don’t discuss the procedures used during bed–chair transfer. Was this training or simply sitting someone on a wheelchair for staying in a more vertical position? Please, provide more information on that in your article.

REPLY: We changed the “recommendation” in “suggestion” for further research and new rehabilitation protocols for patients at risk. We added more details regarding bed-chair transfer

Introduction

Line 35: Not sure what the authors mean by “causative event”. Traumatic? A single factor? Please, reword.

REPLY: We tried to explain that single or multiple factors are not always present at the symptomatic phase which is a common cause of misdiagnosis or late diagnosis.

Line 47: “rulling out other causes”. Please provide some examples of other potential causes as you are discussing differential diagnosis regarding that clinical presentation.

REPLY: We added a list of possible potential causes of leg pain

Case report

Line 68: Not sure what the authors mean by “familiarity”. Do you mean “family history”?

REPLY: Yes, we changed the term as suggested.

Line 71: Please add the score of the CRS-r.

REPLY: CRS-r values have been added

Line 76: Please add “(enoxaparin)” after “low molecular weight heparin”.

REPLY: We specified the molecule

Line 79: Regarding “respiratory rehab” you already state “rehabilitation” at the beginning of the sentence. What aspects of respiratory rehab are relevant in this concern? Respiratory muscles training? Other? Please, be more specific. In the same line, what are the informed consent for? Rehab? This article?

REPLY: We specified the respiratory program for this patient in order to better present our hypothesis on the factors predisposing to hematoma in an anticoagulated patient. In particular, we highlighted the role of prono-supination and limb passive movements in promoting muscle “overload”. Again, we removed the sentence for informed consent.

Line 80: Please provide the new CRS-r score.

REPLY: Done

Lines 97-98: Many readers may not be familiar with the acronyms CPK, INR, aPTT, vn. Please provide the definitions.

REPLY: We specified the acronyms as you suggested.

Figures: Many readers may not easily detect abnormalities. Please, highlight them (suggestion: an arrow or a circle around the enlarged muscle)

REPLY: We modified the images for a better understanding as you suggested.

Discussion:

Line 130: I’m a little bit confused with “preserved active and passive movements” because the authors earlier reported that “She was unable to move the right lower limb. The passive hip extension was limited and very painful” (lines 89–90). Can the authors please explain?

REPLY: Following the reviewer’s suggestion, we better explain this important point. She was able to move the legs, but very intense pain was evoked at the very first degrees of motion, so she was not able to perform our task.

May physical examination predict such event reported in the article? A certain physical performance level should be previously achieved? Any recommendations/hypotheses?

REPLY: The aim of this paper is to raise awareness in this clinical scenario and to stimulate further debate. There is not enough literature to help the clinician with a predictive physical examination model, but a lot of risk factors have been proposed in the last few years.

Evidence has been provided for the acute setting emergency department but also in the intensive care unit and neurosurgery. Cases have been described after a specific physical performance (hematoma in a golf player taking warfarin)(Choa and Lim. Iliopsoas haematoma: an uncommon differential diagnosis for groin pain. Hong Kong Journal of Emergency Medicine 2011), and without physical performance (Artzner T, Clere-Jehl R, Schenck M, et al. Spontaneous ilio-psoas hematomas complicating intensive care unit hospitalizations. PLoS One. 2019; doi: 10.1371/journal.pone.0211680).  We can assume that an active or a passive movement can cause shear stress on the muscle, thus suggesting that movement per se, and not the physical performance, might trigger the hematoma in patients at risk. 

A different rehabilitation approach for a certain period could reduce the risk: a) passive transfer to the wheelchair; b) handling the limbs of the patient with a reduced range of motion to reduce muscle strain or overload; c) earlier suspension of anticoagulant for deep vein thrombosis prophylaxis; or d) drugs predisposing for bleeding. Moreover, it could be of interest to review the literature on psoas and iliacus hematoma to find a window time in which the hematoma can develop after an acute event for a tailored rehabilitation program for patients at high risk.

Reviewer 2 Report

Dear authors,

I find your manuscript very interesting. I would like to propose some changes:

INTRODUCTION

The authors present a case of haematoma in a pregnant woman. In this regard, there are unanswered questions in the introduction: Is this condition common in pregnant women? Why does it occur?

CASE REPORT

The single case methodology is based on taking constant measurements of the chosen variable. Usually, two phases are established: FAse A, before treatment, FAse B during treatment. This is known as Model A-B. Which variable has been chosen and the results of the measurements?

In the description of the clinical situation there is hardly any information about what the cognitive situation of the patient was at the beginning of the rehabilitation treatment and at the end.

DISCUSSION

The discussion should include the debate on the results obtained after continuous measurement of the selected variables.

Best regards

Author Response

INTRODUCTION

The authors present a case of haematoma in a pregnant woman. In this regard, there are unanswered questions in the introduction: Is this condition common in pregnant women? Why does it occur?

REPLY: The patient had a cerebellar hemorrhage during pregnancy in the 7th month. Stroke in pregnancy is uncommon. Complications have been detected in 9-26 cases per 100,000, and deliveries with hemorrhage account for up to 38% of cases (Jaigobin C, Silver FL. Stroke and pregnancy. Stroke 2000;31:2948–51). The most common cause is arteriovenous malformation, other possible causes are eclampsia, thrombosis of the cerebral vein with hemorrage evolution. In this case, the patient had an iliacus hematoma during the rehabilitation program after a long period since the cesarean delivery. We cannot connect this event to the pregnancy.

CASE REPORT

The single case methodology is based on taking constant measurements of the chosen variable. Usually, two phases are established: FAse A, before treatment, FAse B during treatment. This is known as Model A-B. Which variable has been chosen and the results of the measurements?

REPLY: Outcome measurements are now better described. Basically CRS-r and NRS scores as well as vital signs and lab data. Please see pag 3-4

In the description of the clinical situation there is hardly any information about what the cognitive situation of the patient was at the beginning of the rehabilitation treatment and at the end.

REPLY: We agree with this reviewer. We now include CRS-r scores to better describe the overall clinical changes before and after standard neurorehabilitation treatment.

DISCUSSION

The discussion should include the debate on the results obtained after continuous measurement of the selected variables.

REPLY: following the reviewer’s suggestion we now include a new statement for improving the debate about this topic. 

Round 2

Reviewer 2 Report

Dear authors,
Thank you for making the suggested changes. I have no further comments.

Best regards